# Loop-Mediated Isothermal Amplification (LAMP) for the Diagnosis of Zika Virus: A Review

**DOI:** 10.3390/v12010019

**Published:** 2019-12-23

**Authors:** Severino Jefferson Ribeiro da Silva, Keith Pardee, Lindomar Pena

**Affiliations:** 1Department of Virology, Aggeu Magalhaes Institute (IAM), Oswaldo Cruz Foundation (Fiocruz), 50670-420 Recife, Brazil; jeffersonbiotecviro@gmail.com; 2Leslie Dan Faculty of Pharmacy, University of Toronto, Toronto, ON M5S 3M2, Canada; keith.pardee@utoronto.ca

**Keywords:** diagnostic, point-of-care, loop-mediated isothermal amplification (LAMP), Zika virus (ZIKV)

## Abstract

The recent outbreak of Zika virus (ZIKV) in the Americas and its devastating developmental and neurological manifestations has prompted the development of field-based diagnostics that are rapid, reliable, handheld, specific, sensitive, and inexpensive. The gold standard molecular method for lab-based diagnosis of ZIKV, from either patient samples or insect vectors, is reverse transcription-quantitative polymerase chain reaction (RT-qPCR). The method, however, is costly and requires lab-based equipment and expertise, which severely limits its use as a point-of-care (POC) tool in resource-poor settings. Moreover, given the lack of antivirals or approved vaccines for ZIKV infection, a POC diagnostic test is urgently needed for the early detection of new outbreaks and to adequately manage patients. Loop-mediated isothermal amplification (LAMP) is a compelling alternative to RT-qPCR for ZIKV and other arboviruses. This low-cost molecular system can be freeze-dried for distribution and exhibits high specificity, sensitivity, and efficiency. A growing body of evidence suggests that LAMP assays can provide greater accessibility to much-needed diagnostics for ZIKV infections, especially in developing countries where the ZIKV is now endemic. This review summarizes the different LAMP methods that have been developed for the virus and summarizes their features, advantages, and limitations.

## 1. Introduction

Zika virus (ZIKV) is a mosquito-borne virus from the genus *Flavivirus* in the family *Flaviviridae* and consists of two genetically distinct lineages: Asian and African [1,2]. Other notable viruses within this genus include dengue virus (DENV), Japanese encephalitis virus (JEV), West Nile virus (WNV), and yellow fever virus (YFV) [3]. Like most flaviviruses, the ZIKV is an enveloped virus with a capsid 50 nm in diameter and an RNA genome of approximately 11 Kb in length. The genome is translated as a single long open reading frame (ORF) and encodes ten proteins. This includes seven non-structural proteins (NS1, NS2A, NS2B, NS3, NS4A, NS4B, NS5), which mediate viral replication for synthesis of new viral particles, and three structural proteins (C, prM, and E), which comprise the capsid and play a key role in host immune evasion [4,5].

First identified in 1947 in the Zika Forest of Uganda, ZIKV infections in humans remained sporadic for 60 years, with very few cases reported, until April 2007 when the ZIKV caused an outbreak on Yap Island, Federated States of Micronesia [6,7,8]. In 2013, the ZIKV was identified in French Polynesia and spread rapidly across the Pacific, including New Caledonia and Cook Islands [9,10,11]. Coincidentally, during those outbreaks, the link between Guillain–Barré syndrome (GBS) and ZIKV was reported, raising concerns about the neurological tropism of the virus [12]. In May 2015, the first case of ZIKV infection was reported in Brazil [13] and the virus rapidly spread throughout the country and much of Latin America, causing the largest recorded epidemic of the virus to date [14]. The Brazilian epidemic raised great international concern because of severe birth defects, including microcephaly, in neonates born to mothers infected by ZIKV during pregnancy [3,15,16].

ZIKV is transmitted mainly through the bite of infected mosquitoes from the genus *Aedes*, although other vectors may also be involved in the transmission [17,18,19,20,21]. Additionally, other routes of ZIKV transmission have been identified, including blood transfusions, transplacental, perinatal, and sexual intercourse [22,23,24]. ZIKV infection usually causes a self-limiting and a mild illness, where the majority of cases are asymptomatic and, when present, symptoms include fever, headache, rash, conjunctivitis, and arthralgia [25]. In regions where there is a circulation of other arboviruses, such as DENV and chikungunya (CHIKV), the clinical diagnosis of ZIKV infection becomes extremely difficult because of common symptoms. Therefore, laboratory-based molecular diagnosis is of fundamental importance to correctly identify the etiologic agent [3,26].

Given the lack of approved vaccines and antivirals against ZIKV, a rapid and reliable point-of-care (POC) diagnostic test for detection of ZIKV is urgently required for control and prevention measures and to increase the diagnostic capacity of ZIKV-affected, mainly in low-resource areas [27,28]. ZIKV infection is diagnosed in the laboratory by nucleic acid amplification tests or serological methods, including enzyme-linked immunosorbent assays (ELISA), plaque reduction neutralization tests (PRNT), and lateral flow assays (rapid tests) [6,7,29,30,31,32]. Currently, RT-qPCR is considered the gold standard method to detect ZIKV from patient and mosquito samples [6,7,29]. Although RT-qPCR provides high-quality results, the test requires extensive sample preparation, RNA extraction, expensive equipment, and technical expertise to run and interpret the amplification of the viral RNA. Moreover, available serological methods are prone to produce false-positive results due to cross-reaction with other flaviviruses in circulation, such as DENV, and are therefore of limited value [6,7,33].

Loop-mediated isothermal amplification (LAMP) is a powerful alternative POC assay for the virus as it allows rapid, robust, and simple amplification of nucleic acid targets at a single and fixed temperature [34]. The assay has many advantages over RT-qPCR, including rapidity, low cost, high sensitivity, and high specificity. LAMP results can also be easily read with the naked eye through color-based reporters that can be added to the reaction mixture [35]. Importantly, the simple, single-temperature incubation allows LAMP reactions to be performed without expensive equipment, directly in the field [36]. Since the 2015 emergence of the ZIKV in Brazil, many LAMP assays have been developed for diagnosis by research groups across the world [27,37,38,39,40,41,42,43,44,45]. These diagnostic platforms based on LAMP have proven to be specific, sensitive and inexpensive POC tools that can be applied even in resource-limited regions of the world. Here we review the development and application of LAMP methods for the diagnosis of ZIKV and explore the next steps to bring this assay into mainstream use.

## 2. Principles of LAMP Assay

LAMP was first reported in 2000 by Notomi et al. (2000) and has since undergone many adaptations and has been put into practice for the detection of pathogens in samples ranging from animals, plants, and humans [34,46]. Some highlights from adaptations of LAMP include multiplexed LAMP, electrochemical LAMP, and disc-based LAMP [46,47,48]. For RNA viruses, such as ZIKV, DENV, and CHIKV, it is necessary to perform a reverse transcription reaction LAMP (RT-LAMP) [27,39,42,49,50,51,52] which includes the enzymes that first convert RNA → DNA upstream of the LAMP process, unless a LAMP enzyme with both reverse transcriptase and DNA polymerase is used [53].

LAMP is an isothermal nucleic acid amplification method that generally employs a set of four or six different primers, which specifically bind to complementary sequences on the molecular target (Figure 1) [34]. Primer sets for the LAMP assay include the forward outer primer (F3), the backward outer primer (B3), the forward inner primer (FIP), the backward inner primer (BIP), and two other primers designed to accelerate the amplification, including the forward loop primer (LF) and the backward loop primer (LB). With the primers in hand, the LAMP assay can be performed with simple and readily available incubation sources like a water bath or heating block for isothermal heating [35,54,55].

During the initial stages of the LAMP reaction, the inner primers (FIP or BIP) anneal by Watson–Crick complementarity to regions F2c or B2c within the target region. The outer primers (F3 or B3) then hybridizes to region F3c or B3c on the target and initiates the formation of self-hybridizing loop structures by the invasion of strands of DNA sequences already extended from the inner primers, including FIP and BIP. This results in the formation of a dumbbell-shaped DNA, and the same dumbbell structure then becomes a seed for exponential amplification (Figure 1). The addition of loop primers (LB and LB) can accelerate the process for exponential amplification of the target sequence [56]. The final products obtained are a mixture of cauliflower-like structures with multiple loops and concatemers of the DNA with various stem lengths [27,34]. The amplified products can be detected by a variety of methods, including dsDNA binding dyes, turbidity measurement, UV light irradiation, real-time fluorescence, gel electrophoresis, smartphone camera, AC susceptometry, and systems based on lateral flow assays [38,39,44,46,53,57,58].

## 3. Principles LAMP Platforms for ZIKV

Different LAMP platforms have been developed for ZIKV detection in both patient samples and arthropod vectors and are summarized in Table 1 [27,28,37,38,39,40,41,42,44,45,53,57,58,59,60,61,62,63,64]. In this section, we provide an overview highlighting the different LAMP systems, including one-step and two-step protocols (Figure 2).

### 3.1. Step 1: Template Preparation

The LAMP assay can be performed using either a one-step or two-step protocol directly from sample matrices, including serum, semen, urine, saliva, and mosquitoes homogenate. The use of Bst 2.0 polymerase, Bst 3.0 polymerase or OmniAmp polymerase enables the test to be performed in a one-step protocol [38,40,41,42,53,65] (Figure 2). These enzymes have both reverse transcriptase (RT) and DNA polymerase activities at a fixed temperature (50–72 °C), and so can be run isothermally for the direct detection of RNA or DNA targets. It is the robust nature of these enzymes, which are capable of maintaining their activities even in the presence of reaction inhibitors, that enables the assay to be performed in the field without the need to extract the genetic material from the samples [38,40,65]. Based on this advantage, several one-step LAMP assays have been developed for the detection of ZIKV [41,42,53].

Bst 2.0 DNA polymerase is an *in silico* designed enzyme homologue of *Bacillus stearothermophilus* DNA polymerase I. The enzyme contains 5′-3′ DNA polymerase activity and strong strand displacement activity, but lacks 5′-3′ exonuclease activity. Compared to its wild-type homologue, Bst 2.0 DNA polymerase has improved thermostability, salt tolerance, yield, and provides faster amplification [66,67]. Although Bst 2.0 exhibits RT activity [41], it is recommended to also add a dedicated reverse transcriptase to LAMP reactions for more consistent and robust amplification of RNA targets [27,39].

Similar to the Bst DNA polymerase 2.0, Bst DNA polymerase 3.0 is an in vitro derivative of *Bacillus stearothermophilus* Bst DNA polymerase and features further improved amplification reaction properties compared to Bst 2.0. The enzyme has been successfully used for rapid ZIKV detection in both human and mosquito samples [40,53]. Bst DNA polymerase 3.0 exhibits DNA polymerase activity 5′-3′ with either DNA or RNA targets, strong strand displacement activity, improved RT activity compared to Bst DNA polymerase 2.0 and lacks both 5′-3′ and 3′-5′ exonuclease activity. Importantly, Bst DNA polymerase 3.0 displays robust performance even in high concentrations of reaction inhibitors, which has enabled LAMP applications for the ZIKV even without sample pre-treatment or RNA extraction [40,53]. These advantages make Bst DNA polymerase 3.0 an excellent candidate for POC platforms and the rapid detection of ZIKV and other pathogens. OmniAmp is an alternate engineered DNA polymerase, derived from PyroPhage 3173, that has shown strong RT activity and excellent performance in LAMP assays but has yet to be exploited for ZIKV detection [65].

In contrast, the two-step LAMP protocol needs the addition of a reverse transcriptase enzyme together with the DNA polymerase (Figure 2). The use of the two-step LAMP protocol to detect ZIKV has been described by several groups [27,28,57,59]. However, the two-step protocol does carry some disadvantages that make it less practical than a one-step LAMP. The two-step method is more expensive, requires longer times, including additional samples and reagents handling steps, which increases the probability of contamination and error [53]. Taken together, these aspects limit the practical use of two-step LAMP for POC applications.

Recently, Wang et al. (2016) developed an RT-LAMP assay to detect ZIKV using the commercially available Loopamp RNA Amplification Kit (Eiken Chemical Co., Ltd., Japan) [39]. The kit contains a mixture of Bst DNA polymerase and AMV reverse transcriptase for the reaction. Furthermore, the study evaluated three different sample processing methods (proteinase K treatment, alkaline lysis, and boiling lysis) and compared them with RNA extraction using a commercial kit (QIAmp viral RNA mini kit). They showed that the RT-LAMP assay using proteinase K treatment had similar sensitivity compared to RNA extracted using the commercial kit and better than the other two methods.

In another study, Yaren et al. (2017) reported a multiplexed system based on RT-LAMP capable of detecting ZIKV, DENV, and CHIKV in human samples and mosquito homogenate [27]. Their assay was able to detect ZIKV directly in urine and plasma without RNA extraction or pretreatment. In contrast, for reactions using mosquito samples, it was necessary to treat samples with ammonia and ethanol before applying them to the LAMP reaction mixture [27].

### 3.2. Step 2: Practical Implementation

With the fundamentals of LAMP established, we now turn our attention to the practical aspects of designing and running reactions. Most LAMP reactions are carried out in plastic microtubes using a mixture of a buffer, Mg^2+^ ions, enzyme, dNTP, which are generally supplied as part of the kit. This is combined with custom primers and the sample of interest that may contain the target genome. The design of ZIKV primers begins with performing multiple alignments of ZIKV and other arbovirus sequences to identify unique and conserved regions within the ZIKV genome that do not share homology with the related flaviviruses. After choosing the target region, the design of LAMP primers can be carried out using online software such as PrimerExplorer (https://primerexplorer.jp/e/), LAMP Designer Optigene (www.optigene.co.uk/lamp-designer/) or Premier Biosoft (http://www.premierbiosoft.com/isothermal/lamp.html). LAMP primers for ZIKV have been successfully designed to target the genes encoding NS1, NS2A, NS4A, NS5, the capsid and envelope proteins of the ZIKV genome [37,38,39,41,45,63].

In establishing a new LAMP diagnostic assay, it is important to optimize the protocol and include the proper controls to limit the risk of false positives or non-target amplification [40]. Variables to consider in LAMP optimization include Mg^2+^ concentrations, dNTP mix concentrations, primers, time reaction, and temperature reaction, which all have a key role in producing reliable results [40,42,44,64]. Lower LAMP temperatures and excessive free Mg^2+^ ions are associated with non-target amplification [27], and, indeed, previous studies have reported false-positive results in RT-LAMP assays [40,41,44,64].

These parameters have recently been explored by Lee et al. (2016) in their work that optimized RT-LAMP for a ZIKV lateral flow assay (LFA) using Bst 3.0 DNA polymerase [40]. In doing so, they found that optimization of Mg^2+^, dNTP, and the reaction time could provide detection of the ZIKV RNA in 30 min without non-target amplification. In a similar work, Kaarj et al. (2018) optimized an RT-LAMP assay combined with a paper microfluidic chip and smartphone to detect ZIKV in only 15 min. Varying only reaction temperature, they found that no amplification occurred at 70 °C, while non-specific amplification in non-template control (NTC) occurred at the sub-optimal temperature of 65 °C, which, as mentioned above, is known to allow off-target amplification. However, by placing the reaction at 68 °C, the target template was successfully amplified from samples without false positives [44].

Neves et al. (2019) developed a simple system based on an RT-LAMP assay using Bst DNA polymerase 3.0 without RNA extraction or sample pre-treatment [64]. They reported that the reaction time was the most important factor to avoid non-target amplification. Evaluating four reaction durations (5, 10, 15, and 20 min), they found that 10-min incubations were sufficient to detect ZIKV from serum samples, whereas 20-min incubations resulted in false-positive amplifications in non-template control (NTC) using water [64].

### 3.3. Step 3: LAMP Modes of Output

An incredible array of outputs has been designed to allow users to detect positive LAMP reactions. For ZIKV alone, this includes colorimetric detection using fluorescent dyes, UV light irradiation, agarose gel electrophoresis, turbidity, real-time fluorescence, smartphone, lateral flow assay, and AC susceptometry. Below we review each of these output modes, beginning with the fluorescent dyes calcein and SYBR green I dyes, which are some of the most widely used for the detection of LAMP amplicons. Here as target sequences are amplified; the dye binds becoming fluorescent and creating an optical readout of LAMP-mediated detection [42,53,64]. Although it presents adequate sensitivity, SYBR green I could inhibit LAMP reaction if the dye is added during the set-up of the LAMP reaction mixture, and when added after incubation, there is a real risk of cross-contamination between LAMP reaction tubes [68]. Recognizing this challenge, we set out to develop a method to address the issue of cross-contamination following incubation, which is a key limitation of the LAMP assay [53,68]. A few studies have reported closed tube methods to minimize cross-contamination in the LAMP result [69,70,71]. However, our approach was to pre-load reaction tube lids with 1 μL of 1:10 dilution of SYBR green I dye to the center of the tube caps before the reaction. By simply inverting the reactions after isothermal incubation, the SYBR green I mixes with the sample, allowing the results to be observed immediately by eye based on the change in color [53].

As mentioned, calcein has also been used to visualize LAMP products in assays for the detection of ZIKV [39]. Calcein dye is a metal ion binding fluorophore, which can be added prior to incubation, where it forms a complex with magnesium (Mg^2+^ obtained from LAMP reaction). Here the positive generation of LAMP products leads to a change in color from orange to green, which can be visualized by eye under natural light or UV irradiation. ZIKV-negative samples are indicated by an orange color, whereas ZIKV-positive samples are indicated by a green color. More recently, several research groups have reported the use of other dyes for end-point reads of LAMP assays, with potential applications for ZIKV. These include malachite green dye [72,73], hydroxynaphthol blue dye [74,75], evagreen dye [76,77], Goldview II dye [78], Gelred dye [79,80], SYTO fluorescent dye [81,82], and berberine dye [83].

Detection of amplified products in LAMP assays can also be accomplished by agarose gel electrophoresis. Here positive reactions generate ladder-like patterns, which correspond to the series of inverted repeats of the amplified target gene and cauliflower-like structures obtained from LAMP products reactions. This has been a popular method of detection for LAMP-based ZIKV assays [27,39,40,42,44,53,64]. However, visualization of LAMP amplicons by gel electrophoresis is time-consuming and carries an increased risk of cross-contamination between samples due to the massive generation of DNA amplicons during the LAMP assay [46,84].

Turbidity offers an alternate approach to tracking LAMP-based detection. Using a turbidimeter or an absorbance-based (650 nm) plate reader, the optical density of LAMP reactions can be monitored for a white precipitation that is caused by the generation of magnesium pyrophosphate, Mg_2_P_2_O_7_ (product generated during LAMP). Recently, turbidity-based LAMP was used to detect the presence of ZIKV in serum samples [39,41]. In the case of the absence of turbidimeter, a brief centrifugation of the LAMP reaction can be used to pellet the white precipitation for detection by eye under natural light [85].

Coupling the LAMP assay to real-time fluorescence has been used for detection of ZIKV in several applications, including from patient and mosquito samples [27,28,41,57,60]. Although it is a useful strategy, real-time fluorescence monitoring requires costly instrumentation that is incompatible with use in remote and low-resource areas where containment and surveillance are critically required. Moreover, some studies reported the use of ZIKV probes in the LAMP reaction, which limits the application in low-resource countries due to the high cost and the need for expensive equipment for data acquisition and analysis of results [27].

LAMP-based assays have also started to pair with portable companion hardware like the smartphone for reading ZIKV assay results [44,58,60]. Using this approach, Ganguli et al. (2017) developed a microfluidic platform using an RT-LAMP smartphone-enabled assay for simultaneous detection of ZIKV, CHIKV, and DENV from whole blood samples. They showed that this system was specific and sensitive for the detection of ZIKV, including the ability to quantitatively measure the target genetic material from samples [60]. A similar RT-LAMP assay paired a microfluidic chip with smartphone-based measurement to simultaneously detect ZIKV, DENV, and CHIKV in blood, urine, and saliva. They used a system involving RT-LAMP reaction coupled with another technology using quenching of unincorporated amplification signal reporters (QUASR), which offers bright signals, multiplexing capabilities, and reduces the detection of false-positives to analyze fluorescence, and the results can be analyzed by fluorescence intensity or by the naked eye [58].

In another study, Kaarj et al. (2018) developed an RT-LAMP assay using a microfluidic chip combined with a pH indicator and a smartphone to detect ZIKV. Using the pH change of productive LAMP reactions, ZIKV positive samples from urine and human plasma could be quantified with the pH-induced color change after only 15 min using the smartphone camera. Moreover, the assay was specific and provided an impressive 1 copy/µL limit of detection [44]. Also, in the context of microfluidic devices, Sabalza et al. (2018) developed an RT-LAMP assay coupled with reverse dot blot analysis (RDB) to detect ZIKV in human saliva. The system was configured as a bench-top isothermal amplification device and was capable of analyzing 24 samples simultaneously and automatically from sample introduction to detection using the RDB technique [63].

AC susceptometry has also been used as a LAMP readout to detect synthetic ZIKV oligonucleotides [38]. The system contains streptavidin-magnetic nanoparticles (streptavidin-MNPs) premixed with LAMP reagents, including the analyte and biotinylated primers. LAMP results are then measured by a portable AC susceptometer, where the changes of the hydrodynamic volume are probed as Brownian relaxation frequency shifts, which can be used for the quantification of ZIKV. The authors evaluated the analytical sensitivity and showed that LAMP was able to recognize 1 attomolar (aM: 10^−18^ moles per liter) synthetic ZIKV oligonucleotide in 20% serum within 27 min and also reported no cross-reactivity with other viruses, including four types of synthetic NS5 genes from relevant arboviruses, YFV, DENV, WNV, and JEV [38].

Several studies have focused on developing lateral flow assays (LFA) coupled with the LAMP method for end-point detection of ZIKV [37,40]. LFA is a favored technique because it is user-friendly, practical for in-field applications, lightweight, portable, and low-cost. One such study by Song et al. (2016), developed a disposable cassette based on RT-LAMP for rapid molecular detection of ZIKV. The instrument-free POC system uses a chemically heated cup to enable testing without the need for electrical power, which greatly reduces instrumentation complexity, processing time, and cost compared to the gold standard method (RT-qPCR). Using leuco crystal violet (LCV) dye, which is visible with the naked eye, they demonstrated that the RT-LAMP assay combined with a microfluidic cassette could detect as few as 5 plaque-forming units (PFU) of the ZIKV in as little as 40 min [37].

In another study, the authors combined the Bst 3.0 DNA polymerase-based RT-LAMP with an LFA to detect ZIKV [40]. This LFA LAMP platform contains an absorbent pad, a test line, a conjugate pad, and a buffer loading pad. Antidigoxigenin and biotin were also affixed at the control line and test line, respectively, and streptavidin-coated gold nanoparticles (AuNPs) were accrued in the conjugate pad. After 30 min of RT-LAMP assay at 72 °C, the resultant products of the assay were rapidly and simply analyzed using 1 µL of the reaction by the LFA test in less than 5 min with a total assay time of 35 min. Taken together, the research we have summarized suggests that the RT-LAMP assay, in combination with any one of these modes of diagnostic reporting, provides a powerful strategy for the development of POC tools for the rapid diagnosis of ZIKV, and, moreover, this system holds great promise for the POC diagnosis of the many other pathogens.

## 4. LAMP Specificity

For clinical validation of a diagnostic test, determination of its analytical and diagnostic specificity is required. Analytical specificity refers to the ability of an assay to unequivocally detect the target pathogen in a sample while remaining negative in the presence of relevant, non-target pathogens. Diagnostic specificity, in turn, refers to the probability that the test will be negative when the disease is absent (true negative rate).

The LAMP assay’s use of four or six primers, which target six or eight distinct regions within a target, respectively conferring a high analytical specificity and selectivity to diagnostics using this approach [34,86]. In the case of LAMP assays for ZIKV detection, it is important that analytical specificity be assessed in the presence of other mosquito-borne arboviruses found in the region. In the case of the most recent ZIKV outbreak in Latin America, this would include the arboviruses DENV 1-4, CHIKV, and YFV [87,88,89,90,91].

High analytical specificity for LAMP-based ZIKV assays has been reported in both mosquito and human samples. In these studies, no detectable cross-reaction was seen against dengue virus (DENV 1-4), chikungunya virus (CHIKV), yellow fever virus (YFV), West Nile virus (WNV), Saint Louis encephalitis (SLEV), Western equine encephalitis virus (WEEV), Venezuelan equine encephalitis virus (VEEV), Rift Valley fever virus (RVFV), Bussuquara virus (BSQV), Langat virus (LGTV), Powassan virus (POWV), Ilheus virus (ILHV), *Plasmodium falciparum*, influenza virus and other pathogens [37,39,41,42,59].

Recently, Guo et al. (2018) developed a real-time fluorescence RT-LAMP assay for ZIKV detection targeting the NS5 gene. In this study, the authors tested the assay’s specificity against eight pathogens: *Klebsiella pneumoniae* ATCC700603, *Acinetobacter baumannii* ATCC19606, *Streptococcus pneumoniae* ATCC49619, *Staphylococcus aureus* ATCC25923, *Streptococcus mitis* ATCC49456, *Pseudomonas aeruginosa* ATCC27853, *Haemophilus influenzae* ATCC49766, and *Escherichia coli* ATCC25922. The results showed excellent specificity for the detection of ZIKV [57]. While exciting, this study evaluated analytical specificity only against bacterial species and not against arboviruses that cause similar symptoms to ZIKV. Moreover, the ZIKV strain used was the historical MR766 ZIKV strain, which is genetically very different from the contemporary epidemic ZIKV strains.

Several investigations have demonstrated that carry-over contamination and nonspecific amplifications are some of the major problems associated with the LAMP assay, producing false-positive results [40,68,92]. In one study, Kaarj et al. 2018 developed a wax-printed paper microfluidic chip utilizing RT-LAMP, whose results could be detected by smartphone imaging. They used a previously published primer set and observed that temperature was the most important factor for a specific and efficient RT-LAMP reaction to ensure reliable results and eliminate non-target amplification. Assay specificity was evaluated only against the Influenza A/H1N1 virus, which precludes generalization of this result to other arboviruses [44].

In another research, a total of 81 diagnostic specimens (serum and plasma) collected from patients in Nicaragua and Brazil was assayed by LAMP and the results compared with RT-qPCR [41]. They performed the experiments in quadruplicate and considered the positive sample based on one or more positive replicates with a defined melting point. From the total number of samples tested, 55 positive samples were determined by the LAMP assay. Of the 55 positive samples, 39 matched the results from the gold standard method for diagnostic of ZIKV, and 16 were false positives. In contrast, of the 26 LAMP assay-negative samples, 12 were false negatives. The analytical specificity of the LAMP assay was tested against RNAs from BSQV, SLEV, Langat virus, LGTV, ILHV, DENV-2, WNV, YFV, and CHIKV.

In this context, we developed a highly specific RT-LAMP assay for the detection of ZIKV in mosquito samples [53]. We evaluated the analytical specificity with seven arboviruses circulating in Brazil, including ZIKV (PE-243), DENV-1 (PE/97-42735), DENV 2 (PE/95-3808), DENV 3 (PE/02-95016), DENV 4 (PE/10-0081), YFV (17DD), and CHIKV (PE2016-480). The study shows that RT-LAMP is very specific for ZIKV detection [53]. Taken together, these results demonstrate the high specificity of the LAMP assay and prove a potential, inexpensive, and accurate molecular detection tool for the detection of ZIKV in arthropod vectors.

## 5. LAMP Sensitivity

Similar to specificity, analytical and diagnostic sensitivities are key parameters for the validation of a diagnostic test. The analytical sensitivity, also known as the limit of detection (LOD) is the lowest amount of the analyte that can reliably be detected in a sample. On the other hand, diagnostic sensitivity refers to the probability that the test will be positive when the disease is present (true positive rate).

To provide perspective on the analytical sensitivity of different LAMP assays for ZIKV, we assembled a table summarizing this measure for all studies using ZIKV-spiked or RNA-spiked samples (Table 1) [27,28,37,38,39,40,41,42,44,53,57,58,59,60,61,62,63,64]. The top-performing ZIKV RT-LAMP method could detect ZIKV RNA standard at as few as 0.02 PFU/test, using RNA prepared by in vitro transcription. For assays that used RNA extracted from ZIKV specimens, top-performing assays provided greater sensitivity than conventional PCR and demonstrated sensitivity similar to RT-qPCR [39]. Regarding reaction time, the RT-LAMP assay was able to detect viral RNA from ZIKV-spiked serum and urine samples (14.5 TCID_50_/mL) in as little as 15 min. However, this last study found that the limit of detection was less sensitive than the RT-qPCR assay for the diagnosis of ZIKV [28].

While some studies have found LAMP to have lower or similar sensitivity for the ZIKV when compared to RT-qPCR [28,39], others have found LAMP to out-perform RT-qPCR sensitivity. Calvert and coworkers demonstrated that the RT-LAMP assay was able to detect ZIKV RNA in serum and urine and showed sensitivity 10-fold higher than RT-qPCR [59]. Zhao and Feng developed a LAMP assay that detected 0.5 × 10^−9^ or 1.12 × 10^−11^ pmol/µl DNA for NS5 or E genes, respectively. This represented a 100-fold greater sensitivity compared with conventional PCR and RT-qPCR [43].

As mentioned, we have recently developed an RT-LAMP assay for the detection of ZIKV in mosquito samples [53]. To calculate the analytical sensitivity of our RT-LAMP assay, we used crude lysate of *Aedes aegypti* spiked with a 10-fold serial dilution of ZIKV ranging from 10^5^ to 10^−7^ PFU without RNA extraction from ZIKV spiked mosquito samples, with positive detection down to 10^−5^ PFU. Additionally, our RT-LAMP assay was also able to detect ZIKV in mosquito samples that had been previously assayed as negative by RT-qPCR with Ct value ranging from 38.6 to 40.3. Taken together, our results suggest that our modification of the RT-LAMP assay could provide sensitivity 10,000 times greater than RT-qPCR for the detection of ZIKV in mosquito samples, thus corroborating previous studies demonstrating that the sensitivity of the LAMP method can be superior to RT-qPCR [42,43,59]. Importantly, there are a number of reasons that might have involved for this variation in sensitivity, including differences in enzymes and research suppliers, primers, detection systems, and type of biological samples.

## 6. LAMP Assay Evaluation Using Clinical Samples

The suitability of LAMP assays for ZIKV detection in clinical samples has been evaluated in several studies using approximately 800 selected samples from patients with suspected arboviruses infection in Brazil, Nicaragua, and the USA [28,41,42,59,61]. For comparison, clinical samples were also assayed using RT-qPCR. The comparative analysis of the ZIKV LAMP assays with RT-qPCR was based on several statistical parameters, such as sensitivity, specificity, positive predictive value, negative predictive value, likelihood ratio positive, likelihood ratio negative, and overall accuracy [59].

In the first of these studies, Kurosaki et al. (2017) evaluated of the ZIKV RT-LAMP assay using 189 specimens (90 plasma/serum and 99 urine) from 120 suspected arbovirus infection cases and simultaneously tested the samples with RT-qPCR technique as a reference assay to detect ZIKV. The results obtained by RT-LAMP assay were concordant with those of the RT-qPCR. However, no statistical diagnostic parameters were calculated based on the results obtained in this work [28].

In another study, Calvert et al. (2017) evaluated of the ZIKV RT-LAMP assay with 178 clinical samples, including 94 serum and 84 urine specimens. The study found the assay to have a sensitivity of 54.5% (95% CI 45.2%–63.5%), specificity of 94.1% (95% CI 85.8%–97.7%), likelihood ratio positive of 9.3% (95% CI 3.8%–23.9%), likelihood ratio negative of 0.5 (95% CI 0.4%–0.6%) and accuracy of 69.7 (95% CI 62.6%–75.9%) for detection of ZIKV relative to RT-qPCR for diagnosis of ZIKV infection. The positive and negative predictive values were calculated based on pre-test probability resulting in approximately 30% for both predictive values [59].

Other previously published papers also evaluated the efficacy of the RT-LAMP for diagnosis of the ZIKV in clinical samples and compared these results with RT-qPCR [41,42,61]. Using 131 clinical samples from suspected patients collected during the acute phase of the disease, including 68 saliva and 63 urine specimens, an RT-LAMP assay was able to detect the ZIKV in 53.4% of the total number of samples. Interestingly, this study showed that ZIKV detection using saliva had a faster time to positivity compared to urine samples [61].

## 7. LAMP for Detection of ZIKV in Mosquito Samples

Surveillance of ZIKV in mosquitoes is an important measure for identifying potential entry points and monitoring virus activity [93]. LAMP has been evaluated for the detection of the ZIKV in mosquito samples by many groups, including ourselves [27,41,42,53]. Yaren et al. reported a diagnostic tool based on RT-LAMP for the detection of ZIKV in infected mosquito samples [27]. However, the need for RNA extraction, complex reagents, such as ammonia, and the LED blue light for visualization of fluorescent signal limits its applications for POC diagnostics in low-resource countries. Other work by Lamb et al. (2018) developed a one-step protocol without RNA extraction that provided results that can be visualized in under 30 min with the naked eye under natural light. We, too, have concluded that the LAMP assay is specific, sensitive, and robust for the detection of ZIKV in experimentally infected *Aedes aegypti* [42].

Chotiwan and coworkers described a LAMP assay for rapid detection of Asian- and African-lineage Zika viruses without RNA extraction [41]. The authors demonstrated that ZIKV RNA can be detected by the LAMP test using infected C3/36 mosquito cell lines and infected mosquitoes. Results were visualized by turbidity after 1 h of incubation of the sample in a simple heat block.

More recently, we developed a novel RT-LAMP assay to detect ZIKV in mosquitoes samples from in Brazil [58]. In this study, we reported a one-step RT-LAMP assay, specific and very sensitive for the detection of ZIKV in experimentally and naturally infected mosquitoes (*Aedes aegypti* and *Culex quinquefasciatus*) without RNA extraction or reverse transcriptase reaction. Importantly, our assay can be performed in a heat block and the results distinguished between positive and negative samples by color change through the naked eye. In addition, the overall performances of the RT-LAMP assay relative to RT-qPCR for the detection of ZIKV in mosquito samples were performed using 60 mosquito samples collected at the epicenter of the Zika epidemic in Pernambuco state, Brazil [58]. The ZIKV RT-LAMP assay for detection of ZIKV in mosquito samples had a diagnostic sensitivity of 100% (95% CI 88.06% to 100.00%), diagnostic specificity of 91.18 % (95% CI 76.32% to 98.14%), ZIKV prevalence of 46.03% (95% CI 33.39% to 59.06%), positive predictive value of 90.62% (95% CI 76.64% to 96.61%), negative predictive value of 100%, and the overall accuracy of the RT-LAMP test was determined as 95.24% (95% CI 86.71% to 99.01%). Considering these advantages and satisfactory results, RT-LAMP using Bst 3.0 DNA polymerase is more suitable to be used for molecular diagnostics in field settings, mainly in low-resource scenarios.

## 8. Multiplexing

Mosquito-borne viruses such as ZIKV, DENV, and CHIKV have rapidly spread across the world, causing epidemics in several countries in South America and the Pacific region, causing significant global public health concern [14]. In these settings, multiplexed detection of the circulating arboviruses is highly desirable. In the last few years, several platforms based on LAMP have been developed for the simultaneous detection of ZIKV, DENV, and CHIKV [27,58,60]. Although these approaches were able to specifically differentiate the target viruses, they displayed lower sensitivity when compared to monoplex LAMP assays (Table 1).

## 9. Clinical Application and Public Health Perspectives

The current ZIKV epidemic has highlighted key limitations for the POC diagnosis of ZIKV and remains a significant unmet need in low-resource countries. With the fact that ZIKV, DENV, and CHIKV cause similar symptoms and co-circulate in many regions of the world, definitive clinical diagnosis is not possible. Although there are reliable laboratory diagnostic methods for ZIKV infection, there remains a need for lower-cost diagnostic platforms that can be performed in the field and overcomes the drawbacks of the gold-standard RT-qPCR assay. LAMP has proven to be a rapid, easy, accurate, and inexpensive platform for ZIKV diagnostics, especially in remote areas and developing countries which have been heavily affected by this devastating pathogen.

One-step LAMP platforms using enzymes with both reverse transcriptase and DNA polymerase activities, such as Bst DNA Polymerase 3.0 or OmniAmp polymerase are more attractive due to convenience to set up the reactions and less handling of samples, thereby reducing pipetting errors and the risk of contamination. Moreover, protocols that do not required sample pre-treatment or RNA extraction combined with straightforward visualization of results are ideal for POC applications, especially in low-resource settings.

The LAMP for ZIKV has the potential to provide diagnostic performance equal to, or even superior to, RT-qPCR. This type of POC tool can bring decentralization of health care through diagnosis and also have great potential for producing reliable and fast results to assist physicians in decision-making and therapeutic management of patients. The next priorities for the field are to operationalize this capability into robust and simple kits for mainstream use.

## Figures and Tables

**Figure 1 viruses-12-00019-f001:**
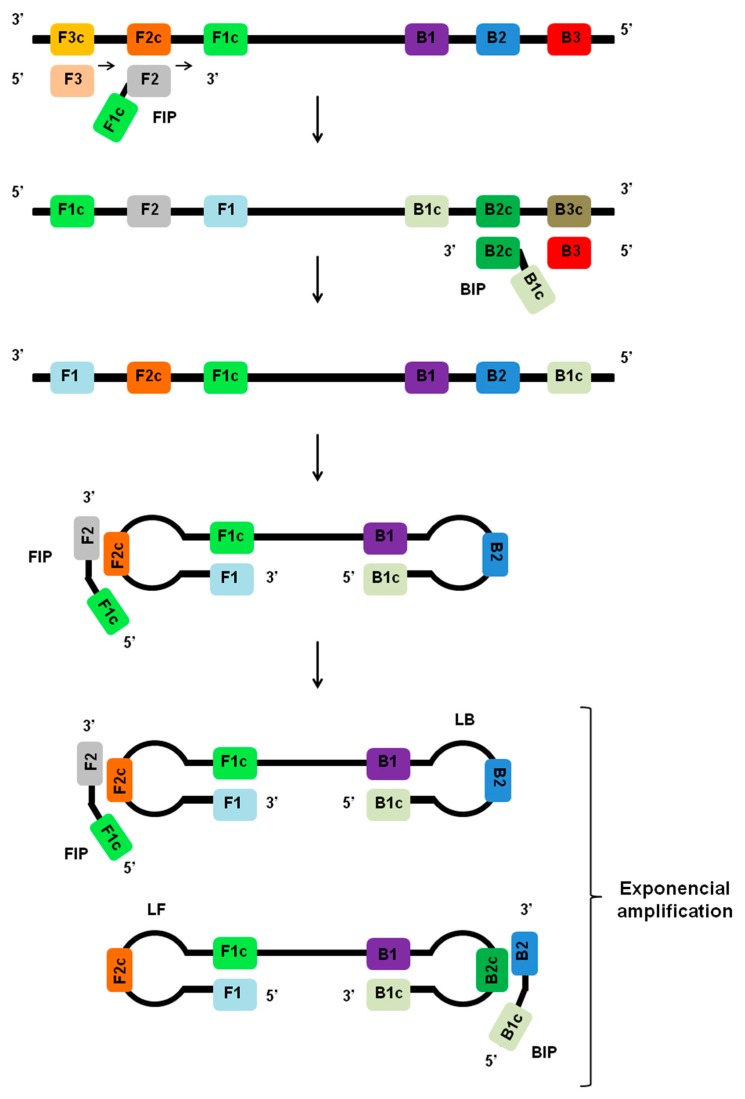
Principles of LAMP assay. During the initial stages of the LAMP reaction, the inner primers (FIP or BIP) anneal to regions F2c or B2c within the target region. The LAMP reaction is initiated by strand invasion by the inner primers, and a strand displacement DNA polymerase extends the primer and separates the target DNA duplex. The outer primers (F3 or B3) then hybridizes to region F3c or B3c on the target and initiates the formation of self-hybridizing loop structures by the strand invasion. This results in the formation of a dumbbell-shaped DNA, which becomes a seed for exponential amplification. As a result of this process, various sized structures consisting of stem-loop DNA with various stem lengths and various cauliflower-like structures with multiple loops are formed. The addition of loop primers (LF and LB) can accelerate the process.

**Figure 2 viruses-12-00019-f002:**
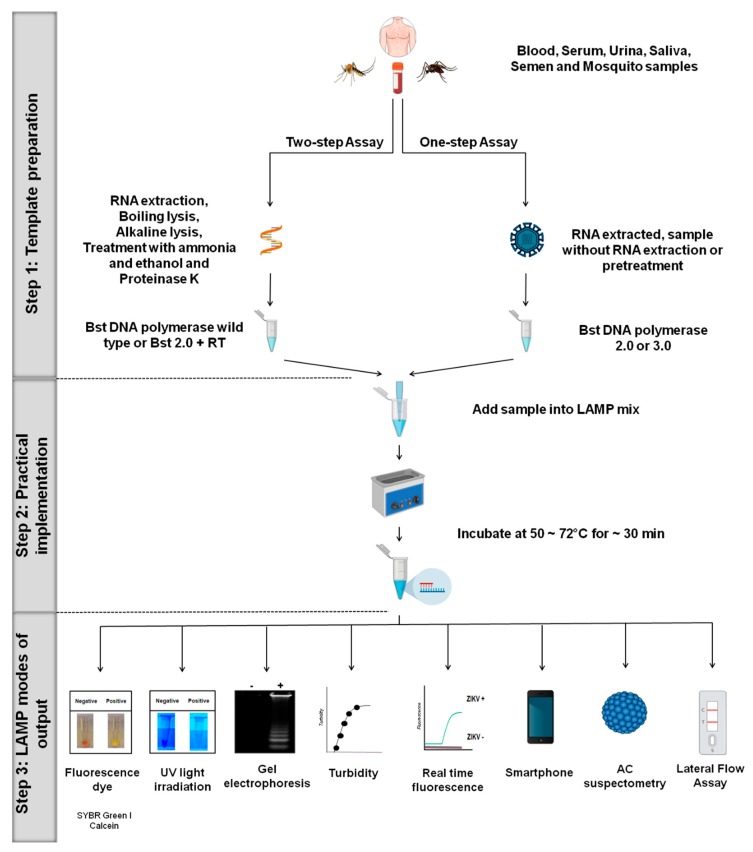
LAMP platforms for ZIKV. Overview of the different LAMP systems for ZIKV, including one-step and two-step protocols. In general, LAMP assays include variations in one of the major three steps: (**1**) Template preparation, (**2**) practical implementation, and (**3**) LAMP modes of output.

**Table 1 viruses-12-00019-t001:** LAMP platforms for Zika virus (ZIKV) diagnostics.

Procedure	Samples	ZIKV Strains Used	Target Region of the Primers	Analytical Sensitivity (Limit of Detection)	Pretreatment or Need for RNA Extraction from the Sample	Validation with Clinical Samples	Reference
One-step RT-LAMP Detection: development of a tape based on lateral flow assay (LFA)	Human blood spiked with RNA ZIKV	MR 766 Uganda (AY632535)	Envelope Protein	10° ZIKV RNA Copy	No	No	[40]
Development of a microfluidic cassette based on RT-LAMP Detection: through the naked eye with the aid of leuco crystal violet (LCV)	Saliva spiked with ZIKV	MEX 2-81 (Mosquito/2016/México)	Envelope Protein	5 PFU	Yes	No	[37]
One-step LAMP Detection: development of LAMP protocol coupled with AC susceptometry	Human serum spiked with synthetic oligonucleotides of ZIKV	Synthetic oligonucleotides of ZIKV	NS5	1 aM	No	No	[38]
One-step RT-LAMP Detection: turbidity monitoring coupled to RT-qPCR. Through the naked eye with calcein and UV light	Saliva, urine, and serum experimentally infected with ZIKV	ZIKV_SMGC-1 China (KX266255)	NS1	0.02 PFU/test	Yes	No	[39]
One-step RT-LAMP Detection: through the colorimetric technology	Clinical samples including urine and human serum	Puerto Rico (PRVABC 59)	Envelope Protein	1.2 RNA copies/μL	Yes	Yes	[59]
One-step RT-LAMP Detection: monitoring of fluorescence coupled to RT-qPCR platform and through the naked eye	Clinical samples including serum, plasma, and human semen. In addition to blood, plasma, saliva, urine, semen and *Aedes* mosquito spiked with ZIKV	Puerto Rico (PRVABC59), P6-740 (HQ234449), 41525 (KU955591) and MR 766 Uganda (AY632535)	NS2A	0.43 PFU	No	Yes	[41]
Development of a Trioplex assay to detect ZIKV, DENV, and CHIKV Detection: development of RT-LAMP protocol coupled with a smartphone for detecting fluorescence	Human blood spiked with ZIKV	Puerto Rico (PRVABC59),	NS1	1.56 PFU/mL	Yes	No	[60]
Development of a platform based on RT-LAMP assay Detection: fluorescence monitoring coupled to RT-qPCR platform	Clinical samples, including serum and human urine. In addition to human serum and urine spiked with ZIKV	PRABC59, 976 Uganda, ArD157995, P6-740, CPC-0740*, 41525-DAK*	Envelope Protein	14.5 TCID_50_/mL	Yes	Yes	[28]
Development of a Trioplex assay to detect ZIKV, DENV, and CHIKV Detection: monitoring of fluorescence coupled to quenching of unincorporated amplification signal reporters (QUASR) and smartphone	Blood, urine, and saliva spiked with ZIKV	Puerto Rico (PRVABC59), Honduras (R103451), and Brazilian strain;	NS5	2 PFU/mL	Yes	No	[58]
Development of a Trioplex kit trial to detect ZIKV, DENV, and CHIKV Detection: monitoring of fluorescence coupled to RT-qPCR platform and through the naked eye	Urine, saliva, plasma and *Aedes* mosquito spiked with ZIKV	Puerto Rico, (PRVABC59, KU501215.1)	NS5	~ 0.71 PFU equivalent viral RNAs	Yes	No	[27]
One-step RT-LAMP Detection: through the naked eye with SYBR Green I	Urine, serum, and infected mosquito samples	MR 766 Uganda (AY632535); MEX20 (ZK-HU 0165 P), Puerto Rico (PRVABC 59) and PB81 (H815744)	NS5	1 copy of the genome/rxn	No	No	[62]
Development of a platform based on RT-LAMP Detection: monitoring of fluorescence coupled to RT-qPCR platform	Clinical samples, including saliva and urine. In addition to saliva and urine spiked with ZIKV	(PRVABC59 – NR-50240)	Capsid	2.2 × 10^3^ RNA copies/mL	Yes	No	[61]
Development of a platform based on RT-LAMP Detection: monitoring of fluorescence coupled to RT-qPCR	Viral RNA	MR 766 Uganda	NS5	3.3 ng/μL	Yes	No	[57]
Development of RT-LAMP platform coupled with on paper microfluidic chips Detection: the color change is monitored in situ with a smartphone	Plasma and urine spiked with ZIKV	Purified ZIKV particles	NS5	1 copy/μL	Yes	No	[44]
One-step RT-LAMP Detection: through the naked eye with SYBR Green I	Clinical samples, including human urine. In addition to urine and *Aedes* mosquito spiked with ZIKV	MR 766 Uganda (AY632535), MEX20 (ZK-HU 0165 P), Puerto Rico (PRVABC 59), and PB81 (H815744)	NS5	1 copy of the genome/rxn	No	No	[42]
Development of an RT-LAMP platform coupled with reverse dot blot analysis (RDB) Detection: monitoring of fluorescence and reverse dot-blot for detection	Saliva spiked with ZIKV	Thailand (PLCal_ZV), Puerto Rican (NR-50244), Thailand (NR-50242), Florida (NR5024), and Honduras (NR-50358)	Capsid	2.10^3^ RNA copies /mL	Yes	No	[63]
Development of a LAMP Detection: a calcein/Mn2+ complex chromogenic method and real-time turbidity monitoring	Viral RNA	Recombinant plasmids containing NS5 gene or E (ZIKV strain Natal RGN)	NS5 and envelope protein	0.5 × 10^−9^ pmol/µL DNA for NS5 and 1.12 × 10^−11^ pmol/µL DNA for envelope protein	Yes	No	[43]
One-step RT-LAMP Detection: through the naked eye with SYBR Green I	Mosquito samples including experimentally and naturally infected mosquitoes (*Aedes aegypti* and *Culex quinquefasciatus*)	PE-243 (KX197192)	Envelope protein	10^−5^ PFU	No	Yes	[53]
One-step RT-LAMP Detection: through the naked eye with SYBR Green I	Serum spiked with ZIKV and patient samples including human serum	-		10^−3^ copies of RNA (20 zepto-molar)	No	Yes	[64]
One-step RT-LAMP Detection: turbidity monitoring	Serum and urine spiked with ZIKV	PRVABC59 (KX601168), MRS_OPY_Martinique_Pari_2015 (KU647676), H/PF/2013 (KJ776791), and MR766 (LC002520)	Envelope protein/NS4A	0.17 FFU/mL–2.3 × 10^2^ FFU/mL	Yes	No	[45]

Abbreviations: aM, attomolar; PFU, plaque-forming unit; FFU, focus forming units; TCID_50_, 50% tissue culture infective dose.

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
