# Peer review of "Loop-Mediated Isothermal Amplification (LAMP) for the Diagnosis of Zika Virus: A Review"

_viruses, 2019, doi:10.3390/v12010019_

Round 1

Reviewer 1 Report

In this manuscript, da Silva et al., looked at the potential use of a relatively new technology LAMP (loop-mediated isothermal amplification) as an easy method for rapid detection of Zika virus (ZIKV) infections. This method does not require specialised technology or training and could also be used for other arboviruses. It could change how diagnostics of ZIKV occurs in poor countries or in areas without resources to use specialised techniques like real time PCR.

This is a very comprehensive review of the use of LAMP technology for ZIKV detection. It merits publication as it will interest many readers. Overall the manuscript is well conceived and thorough. However, I do have some comments to improve on the current form. Also, there were some problems with the English language spelling/grammar/structure and should be revised. Some of the issues have been highlighted.

Comments: 

1) I find Fig.1 a little confusing for people who may not be familiar with the method. For example, the last sentence of the figure legend refers to loop primers LB LB but the image depicts LB and LF. Furthermore, it is unclear that during the amplification process that the primers will lead to the production of both the complimentary structure of the original stem loop DNA and one gap repaired stem loop. For reference, one of the best explanations of this method that I have seen was at 

http://premierbiosoft.com/tech_notes/Loop-Mediated-Isothermal-Amplification.html.

The authors may want to review this figure and rectify to enhance clarity.

2) Why in table one, is the choice yes or not. Should it not be yes or no?

3) On line 184: (...) "that do not share homology with the related arboviruses".

I feel that it is .ot only related arboviruses. As mentioned throughout the manuscript, detection of ZIKV is often paired with the detection of other arboviruses co-circulating in a given area. In this instance, ZIKV is often circulating alongside DENV and CHIKV. Appropriate identification is paramount as patients infected with these viruses present similar symptoms. However, CHIKV is not related to ZIKV as it is an alphavirus and not a flavivirus. Perhaps the authors want to add a sentence.

4) On line 233: However, our approach was to pre-loaded reaction tube lids with 1 μL ...

This sentence should read to pre-load.

5) Lines 233-236: However, our approach was to pre-loaded reaction tube lids with 1 μL of 1:10 dilution of SYBR Green I dye during reaction set-up. By simply inverting the reactions after isothermal incubation, the SYBR green I mixes with the sample, allowing the results to be observed immediately by eye based on the change in color.

What do the authors mean by this statement? Do the lids stay open during incubation? If not, how can you assure that the solution does not fall into the tube when capping? Please clarify.

6)  On line 239:  Calcein dye is a metal ion binding fluorophore, which can added prior to incubation ...

Authors should write which can be added.

7) On line 256:  (...) or an absorbance-based (650 nm) plater reader(...)

Please correct to plate reader.

8) Lines 266-267: Moreover, some studies reported the use of ZIKV probes in the LAMP reaction, which limits the application in low-resource countries due to the high cost of the consumables for the LAMP assay.

Is this not true for SyBr Green I dyes? Are they not more costly than other dyes? This was explained just prior to the statement.

9) Line 296: (...) reported no cross-reactivity with other viruses including four types of synthetic NS5 gene from (...)

Please correct to genes

10) Figure 2 is well presented and helps synthesize the content of this review.

11) Line 366: (...)analytical specificity of the LAMP assay was tested against RNAs from BSQV, SLEV, Langat virus, LGTV, ILHV, DENV-2, WNV, YFV, and CHIKV.

These acronyms have not been defined in the text.

12)  Lines 377-378: Similar to specificity, analytical and diagnostic sensitivity is a key parameter for the validation of a diagnostic test.

Please change to sensitivities are key parameters as this refers to both analytical and diagnostic sensitivities.

13) Line 447: However, the need RNA extraction

Please add need for RNA

14)  Line 451: We to have concluded

This should be we too have concluded. I suspect the authors meant to use the adverb not the preposition.

15) Line 488: LAMP have been proven

This should be has proven

16) Line 494: With this established next priorities for the field are to operationalize this capability into robust and simple kits for mainstream use.

Please revise the syntax of this sentence.

17) It would be of interest in the conclusion, if the authors stated their opinion on the best current method of detecting ZIKV in poor countries. There was a lot of interesting information throughout this review. Is there a stand-out method that could be used as a basis for further development areas that may struggle to use the current gold-standard method aka real time PCR?

If not, perhaps they should summarise key features the twould lead to the ultimate refined LAMP method. For example, one that does not require RNA extraction (so one-step LAMP), one that uses easy and cheap detection methods perhaps turbidity or the cheapest dye (probably not smartphone linked), one with proven robust primer set, etc.

Author Response

Reviewer #1:

Comments and Suggestions for Authors

In this manuscript, da Silva et al., looked at the potential use of a relatively new technology LAMP (loop-mediated isothermal amplification) as an easy method for rapid detection of Zika virus (ZIKV) infections. This method does not require specialised technology or training and could also be used for other arboviruses. It could change how diagnostics of ZIKV occurs in poor countries or in areas without resources to use specialised techniques like real time PCR.

This is a very comprehensive review of the use of LAMP technology for ZIKV detection. It merits publication as it will interest many readers. Overall the manuscript is well conceived and thorough. However, I do have some comments to improve on the current form. Also, there were some problems with the English language spelling/grammar/structure and should be revised. Some of the issues have been highlighted.

Comments

1) I find Fig.1 a little confusing for people who may not be familiar with the method. For example, the last sentence of the figure legend refers to loop primers LB LB but the image depicts LB and LF. Furthermore, it is unclear that during the amplification process that the primers will lead to the production of both the complimentary structure of the original stem loop DNA and one gap repaired stem loop. For reference, one of the best explanations of this method that I have seen was at

http://premierbiosoft.com/tech_notes/Loop-Mediated-Isothermal-Amplification.html.

The authors may want to review this figure and rectify to enhance clarity.

            Response: All suggested changes have been made (page 4, line 190-197). Where it reads in the “loop primers LB LB” we actually mean “loop primers LF and LB”. We have corrected that typo and have also improved the description in the figure legend.

2) Why in table one, is the choice yes or not. Should it not be yes or no?  

            Response: Yes, we considered the suggestion made by the reviewer in table 1. All suggested changes have been made (Table 1).

3) On line 184: (...) "that do not share homology with the related arboviruses".

I feel that it is not only related arboviruses. As mentioned throughout the manuscript, detection of ZIKV is often paired with the detection of other arboviruses co-circulating in a given area. In this instance, ZIKV is often circulating alongside DENV and CHIKV. Appropriate identification is paramount as patients infected with these viruses present similar symptoms. However, CHIKV is not related to ZIKV as it is an alphavirus and not a flavivirus. Perhaps the authors want to add a sentence.

            Response: All suggested changes have been made (page 11, line 229).

 4) On line 233: However, our approach was to pre-loaded reaction tube lids with 1 μL.

This sentence should read to pre-load.

            Response: All suggested changes have been made (page 12, line 271).

5) Lines 233-236: However, our approach was to pre-loaded reaction tube lids with 1 μL of 1:10 dilution of SYBR Green I dye during reaction set-up. By simply inverting the reactions after isothermal incubation, the SYBR green I mixes with the sample, allowing the results to be observed immediately by eye based on the change in color.

What do the authors mean by this statement? Do the lids stay open during incubation? If not, how can you assure that the solution does not fall into the tube when capping? Please clarify.

            Response: Our approach is to add 1 mL of SYBR Green I to the center of the tube caps before reaction. By simply inverting the reactions after isothermal incubation, the SYBR green I is mixed with the sample, allowing the results to be observed immediately by eye based on the change in color. All suggested changes have been made (page 12, lines 272-273).

6) On line 239:  Calcein dye is a metal ion binding fluorophore, which can added prior to incubation ...Authors should write which can be added.

            Response: All suggested changes have been made (page 12, line 272).

7) On line 256:  (...) or an absorbance-based (650 nm) plater reader(...)

Please correct to plate reader.

            Response: All suggested changes have been made (page 12, line 294).

8) Lines 266-267: Moreover, some studies reported the use of ZIKV probes in the LAMP reaction, which limits the application in low-resource countries due to the high cost of the consumables for the LAMP assay.

Is this not true for SyBr Green I dyes? Are they not more costly than other dyes? This was explained just prior to the statement.

            Response: SYBR Green I dye is considered a good alternative for use in LAMP assays with great potential for use in remote areas. However, some studies have suggested the use of probes for use in the LAMP assay, which greatly limits its application due to the cost and the need of expensive equipment for data acquisition and analysis (Priye et al. 2017; Yaren et al. 2017). We clarify this information in the manuscript (page 13, lines 305-306).

            Priye, A.; Bird, S. W.; Light, Y. K.; Ball, C. S.; Negrete, O. A.; Meagher, R. J., A smartphone-based diagnostic platform for rapid detection of Zika, chikungunya, and dengue viruses. Sci Rep 2017, 7, 44778.

           Yaren, O.; Alto, B. W.; Gangodkar, P. V.; Ranade, S. R.; Patil, K. N.; Bradley, K. M.; Yang, Z.; Phadke, N.; Benner, S. A., Point of sampling detection of Zika virus within a multiplexed kit capable of detecting dengue and chikungunya. BMC Infect Dis 2017, 17, (1), 293.

9) Line 296: (...) reported no cross-reactivity with other viruses including four types of synthetic NS5 gene from (...)

Please correct to genes

            Response: All suggested changes have been made (page 13, line 334).

10) Figure 2 is well presented and helps synthesize the content of this review.

            Response: We appreciate the reviewer's comment.

11) Line 366: (...) analytical specificity of the LAMP assay was tested against RNAs from BSQV, SLEV, Langat virus, LGTV, ILHV, DENV-2, WNV, YFV, and CHIKV.

These acronyms have not been defined in the text.

Response: These acronyms have been defined (page 15, lines 417-421).

12) Lines 377-378: Similar to specificity, analytical and diagnostic sensitivity is a key parameter for the validation of a diagnostic test.

Please change to sensitivities are key parameters as this refers to both analytical and diagnostic sensitivities.

Response: All suggested changes have been made (page 16, line 461).

13) Line 447: However, the need RNA extraction

Please add need for RNA

Response: All suggested changes have been made (page 17, line 527).

14) Line 451: We to have concluded

This should be we too have concluded. I suspect the authors meant to use the adverb not the preposition.

Response: All suggested changes have been made (page 17, line 531).

15) Line 488: LAMP have been proven

This should be has proven

Response: All suggested changes have been made (page 18, line 568).

16) Line 494: With this established next priorities for the field are to operationalize this capability into robust and simple kits for mainstream use.

Please revise the syntax of this sentence.

Response: All suggested changes have been made (page 18, lines 580-581).

17) It would be of interest in the conclusion, if the authors stated their opinion on the best current method of detecting ZIKV in poor countries. There was a lot of interesting information throughout this review. Is there a stand-out method that could be used as a basis for further development areas that may struggle to use the current gold-standard method aka real time PCR?

If not, perhaps they should summarise key features the twould lead to the ultimate refined LAMP method. For example, one that does not require RNA extraction (so one-step LAMP), one that uses easy and cheap detection methods perhaps turbidity or the cheapest dye (probably not smartphone linked), one with proven robust primer set, etc.

Response: We appreciate the reviewer's comment. We have added a paragraph addressing the points raised by the reviewer (page 18, lines 571-576).

Reviewer 2 Report

The article “Loop-Mediated Isothermal Amplification (LAMP) for the Diagnosis of Zika Virus: A Review” by da Silva and Pena follows the development and usefulness of various LAMP methods to detect flaviviruses, especially Zika virus at a low cost compared to conventional detection methods. The review is well written, organized and covered most of the relevant topics. The following need to be addressed before consideration of this review for publication:

The authors quote a “manuscript in preparation” from them in many places in this manuscript which is not acceptable. Authors should wait for the quoted manuscript to be published before including in this review or include the data to substantiate their claim. Spelling/grammar errors Line 65; “costly, expensive” means the same Line 239; “which can added” Line 256; “plater reader” Line 269; “started to paired” Line 417; “evaluated of the” Line 448; “visualized” Line 460; “transcriptase reverse” Line 484; “one a” Line 486; “diagnostic”

3.The authors claim the LAMP is cost effective to detect flaviviruses. A real or predicted cost comparison between LAMP based and conventional methods of flavivirus detection at POC should be included.

Author Response

Reviewer #2:

Comments and Suggestions for Authors

The article “Loop-Mediated Isothermal Amplification (LAMP) for the Diagnosis of Zika Virus: A Review” by da Silva and Pena follows the development and usefulness of various LAMP methods to detect flaviviruses, especially Zika virus at a low cost compared to conventional detection methods. The review is well written, organized and covered most of the relevant topics. The following need to be addressed before consideration of this review for publication:

1) The authors quote a “manuscript in preparation” from them in many places in this manuscript which is not acceptable. Authors should wait for the quoted manuscript to be published before including in this review or include the data to substantiate their claim.

Response: We appreciate the reviewer's comment. All information regarding our manuscript in preparation has been removed.

2) Spelling/grammar errors Line 65; “costly, expensive” means the same Line 239; “which can added” Line 256; “plater reader” Line 269; “started to paired” Line 417; “evaluated of the” Line 448; “visualized” Line 460; “transcriptase reverse” Line 484; “one a” Line 486; “diagnostic”.

Response: All suggested changes have been made.

3) The authors claim the LAMP is cost effective to detect flaviviruses. A real or predicted cost comparison between LAMP based and conventional methods of flavivirus detection at POC should be included.

Response: We were not able to find a published article comparing the costs of LAMP for flaviviruses with qRT-PCR. We have done cost comparison and the estimated reagent cost for a LAMP assay is US$ 0.26 as opposed to US$ 10.75 for qRT-PCR. We will submit our manuscript shortly reporting this and other results obtained with humans samples. In this review, we do cite Mori and Notomi, 2009 which mentions the cost advantages of LAMP assays.

Mori, Y.; Notomi, T., Loop-mediated isothermal amplification (LAMP): a rapid, accurate, and cost-effective diagnostic method for infectious diseases. Journal of infection and chemotherapy : official journal of the Japan Society of Chemotherapy 2009, 15, (2), 62-9.

Reviewer 3 Report

The authors present a review of the application of LAMP assays for Zika virus nucleic acid detection. The value of the review includes the presentation of basic parameters and current methods for LAMP assays that are applicable to POC applications for other flaviviruses as well as other infectious agents.

On lines 86-89, the authors state: “For RNA viruses, such as 86 ZIKV, DENV and CHIKV, it’s necessary to perform a reverse transcription reaction LAMP 87 (RT-LAMP) [27, 39, 42, 48-51] which includes the enzymes that first convert RNA -> DNA upstream 88 of the LAMP process [52]. While RNA must be reverse transcribed for amplification, the statement suggests a need for a separate reverse transcription step with a reverse transcriptase enzyme.  The authors later describe, in detail, the inherent RT activities of the various Bst enzyme preparations.  Perhaps this aspect could be included in this initial introduction of the topic or at least indicate that reverse transcription and amplification are likely coincident with these enzymes. The reference numbers in Table 1 are not the correct references.For example, reference 1 for a One-step – RT-LAMP with lateral flow assay is Haddow et al, “Characterization of Zika virus strains….”.  I assume that the table reference numbers were placeholders to be changed to the final reference numbers after formatting.  Please correct. Without the reference numbers in table 1 it is impossible to evaluate the source material, but the numbers in the limit of detection column are hard to interpret and, in a few cases, are theoretically impossible, eg. 1 copy of genome /reaction; neither RT-PCR or LAMP can detect a single virus RNA copy in a reaction, with statistical confidence.The same is the case for 1 aM or 20 zeptomolar concentrations (~12,000 copies per liter or 0.012 copies per µL).  These results should be suspect, as it is difficult to detect less than one of a thing in a reaction.

In other cases, a number of biological materials are listed but only a single limit of detection is provided without identification of the source material.  There are wide variations in the LODs dependent on direct LAMP reactions without RNA isolation or on purified RNA.

This applies in sections 4, 5 and 6 as well, where comparisons between LAMP and RT-PCR are referred to, but without information regarding the source material, crude or purified. It is not made clear, in describing specific studies, whether LAMP assays were performed on crude preparations and compared to RT-PCR on purified RNAs; false positives and negatives drop precipitously when purified RNA is used for LAMP.  And, despite being the gold standard, RT-PCR has false negatives, when subsequent serology results are available for clinical samples.  A percentage of “false positive” LAMP results are ultimately true positives by serology and the matched RT-PCR results were actually false negatives.

Therefore, the descriptions of studies chosen as examples for each of these sections should be expanded with greater details on the experimental conditions used in such direct comparisons.

Line 342. The use of non-target bacterial species as negative controls for ZIKV LAMP assays (ref. 56) is not particularly pertinent to assay specificity.

There are a number of typographical errors and errors of agreement between noun and verb (plural vs. singular forms) throughout the manuscript that should be corrected.

The authors describe data from their unpublished manuscript under almost every heading of the review.  Reviews should not be used as a means of pre-publication or promotion of unpublished data.

Author Response

Reviewer #3:

Comments and Suggestions for Authors

The authors present a review of the application of LAMP assays for Zika virus nucleic acid detection. The value of the review includes the presentation of basic parameters and current methods for LAMP assays that are applicable to POC applications for other flaviviruses as well as other infectious agents.

1) On lines 86-89, the authors state: “For RNA viruses, such as 86 ZIKV, DENV and CHIKV, it’s necessary to perform a reverse transcription reaction LAMP 87 (RT-LAMP) [27, 39, 42, 48-51] which includes the enzymes that first convert RNA -> DNA upstream 88 of the LAMP process [52]. While RNA must be reverse transcribed for amplification, the statement suggests a need for a separate reverse transcription step with a reverse transcriptase enzyme.  The authors later describe, in detail, the inherent RT activities of the various Bst enzyme preparations. Perhaps this aspect could be included in this initial introduction of the topic or at least indicate that reverse transcription and amplification are likely coincident with these enzymes. The reference numbers in Table 1 are not the correct references. For example, reference 1 for a One-step – RT-LAMP with lateral flow assay is Haddow et al, “Characterization of Zika virus strains….”.  I assume that the table reference numbers were placeholders to be changed to the final reference numbers after formatting.  Please correct. Without the reference numbers in table 1 it is impossible to evaluate the source material, but the numbers in the limit of detection column are hard to interpret and, in a few cases, are theoretically impossible, eg. 1 copy of genome /reaction; neither RT-PCR or LAMP can detect a single virus RNA copy in a reaction, with statistical confidence. The same is the case for 1 aM or 20 zeptomolar concentrations (~12,000 copies per liter or 0.012 copies per µL).  These results should be suspect, as it is difficult to detect less than one of a thing in a reaction.

In other cases, a number of biological materials are listed but only a single limit of detection is provided without identification of the source material. There are wide variations in the LODs dependent on direct LAMP reactions without RNA isolation or on purified RNA.

            Response: All suggested changes have been made.

2) This applies in sections 4, 5 and 6 as well, where comparisons between LAMP and RT-PCR are referred to, but without information regarding the source material, crude or purified. It is not made clear, in describing specific studies, whether LAMP assays were performed on crude preparations and compared to RT-PCR on purified RNAs; false positives and negatives drop precipitously when purified RNA is used for LAMP.  And, despite being the gold standard, RT-PCR has false negatives, when subsequent serology results are available for clinical samples. A percentage of “false positive” LAMP results are ultimately true positives by serology and the matched RT-PCR results were actually false negatives.

Therefore, the descriptions of studies chosen as examples for each of these sections should be expanded with greater details on the experimental conditions used in such direct comparisons.

            Response: Table 1 provides the information the reviewer mentioned in detail, including the type of sample and if there were RNA extraction of pretreatment of the samples. We believe that including this information in the text will make this review to lengthy and hampers readability. Thus, we think that providing this information in a table format will be the best way for readers to compare the different studies.

3) Line 342. The use of non-target bacterial species as negative controls for ZIKV LAMP assays (ref. 56) is not particularly pertinent to assay specificity.

Response: All suggested changes have been made (page 15, pages 429-432).

4) There are a number of typographical errors and errors of agreement between noun and verb (plural vs. singular forms) throughout the manuscript that should be corrected.

            Response: All suggested changes have been made.

5) The authors describe data from their unpublished manuscript under almost every heading of the review. Reviews should not be used as a means of pre-publication or promotion of unpublished data.

            Response: We appreciate the reviewer's comment. All information regarding our unpublished manuscript has been removed.